# Rare Intercondylar Distal Femoral Brodie’s Abscess in a 21-Year-Old Man Who Refused Medical Care for Three Years after Initial Symptoms

**DOI:** 10.3390/medicina57060544

**Published:** 2021-05-28

**Authors:** Bogdan Gheorghe Hogea, Jenel Marian Patrascu, Adrian Emil Lazarescu, Louchi El Mehdi, Andrei Daniel Bolovan, Lavinia Maria Hogea, Adrian Cosmin Ilie, Bogdan Corneliu Andor, Jenel Marian Patrascu

**Affiliations:** 1Faculty of Medicine, University of Medicine and Pharmacy ‘Victor Babes’, Nr. 2, 300041 Timisoara, Romania; hogeabg@yahoo.com (B.G.H.); jenel.patrascu@umft.ro (J.M.P.J.); laviniahogea@yahoo.com (L.M.H.); ilie.adrian@umft.ro (A.C.I.); andormed@yahoo.com (B.C.A.); jenelmarianp@yahoo.com (J.M.P.); 22nd Clinic of Orthopedics and Traumatology, County Emergency Hospital ‘Pius Branzeu’, Nr. 2, 300041 Timisoara, Romania; mehdilouchi@gmail.com (L.E.M.); andrei.bolovan30@gmail.com (A.D.B.); 3‘Professor Teodor Sora’ Research Center, U.M.F., Nr. 2, 300041 Timisoara, Romania

**Keywords:** Brodie’s abscess, infection, chronic osteomyelitis

## Abstract

Brodie’s abscess is a rare form of sub-acute osteomyelitis that implies the collection of pus inside bone tissue. The present paper presents an extremely rare case of Brodie’s abscess located in the distal femur in a young male patient who refused medical care for three years and presented directly with spontaneous fistula and septic complications. Laboratory tests also suggested chronic septic alterations. Complex imaging investigations including X-ray (RX), computer tomography (CT) and Magnetic Resonance imaging (MRI) confirmed the diagnosis with characteristic aspects, such as the penumbra sign on the T1 weighted MRI image. Management included aggressive debridement, defect reconstruction, and long-term specific antibiotics according to culture harvested intra-operatively. Evolution was positive with inflammatory blood tests returning to physiological values within four weeks and patient full recovery within six months, without any physical deficits. The novelty aspect found in this case presentation is represented by the long-term natural evolution of this pathology, and the fact that even in these conditions, the Brodie’s abscess did not evolve into a ‘malignant’ septic condition, but remained rather benign until the spontaneous fistula prompted the patient to seek medical care.

## 1. Introduction

Brodie’s abscess is a rare form of sub-acute osteomyelitis that implies the collection of pus inside bone tissue, usually with insidious onset and local clinical findings rather than systemic symptoms [1]. In a series of 20 cases, Stephens et al. found that the majority of cases occurred in the second decade of life, all had local symptoms for six weeks or more, and most cases involved the proximal tibia [2]. Because it is considered a rare pathology, most of the literature published on Brodie’s abscess consists of case presentations, and only a few case series starting with the well-known paper by Boriani [3], which analyzed 181 cases, and the paper of Gillespie [4] analyzing 57 cases of subacute pyogenic osteomyelitis known as Brodie’s abscess. Treatment consists generally of surgery (94%) combined with antibiotherapy (77%) [1]. The present paper presents an interesting case of Brodie’s abscess located in the distal femur of a young male patient.

## 2. Materials and Methods

A 21 year old patient presents to our clinic with pain in the left knee and a fistula present on the posterior aspect of the knee with abundant secretion, local swelling, and good general condition. The patient describes a three-year history of periodic knee pain episodes with insidious onset and progressive evolution that subsided with conservative methods without medical examination and management. The patient refused medical care until the spontaneous occurrence of the fistula, which prompted him to present in our clinic three days after spontaneous drainage. From the anamnestic data we could not identify any history of repeated trauma to the knee area that could be involved in the pathology of Brodie’s abscess [1].

On clinical examination, the range of motion was extremely limited and the clinical signs of infection were all present (calor, rubor, dolor) together with the presence of the fistula with pus secretion present. There was extreme local swelling and continuous pain with no relation to position or movement. The patient was admitted immediately for further investigation.

Blood tests revealed a few significant alterations that consisted of anemia (hemoglobin of 9.2 g/dL [ref 11.5–15]) most certainly caused by chronic septic syndrome, a slight elevation of lymphocytes (42.1% [ref 20–40%]) and monocytes (13% [ref 3.5–9.5%]), a slight lowering of neutrophils (38.4% [ref 40–70%]), and an elevation of eosinophils (6% [ref 0–4%]). Inflammatory tests revealed an elevated RCP (18.72 mg/L [ref 0–3 mg/L]) with normal levels of fibrinogen and procalcitonin, and a normal sedimentation rate.

Before hospital admission, the patient went for a plain X-ray of the knee, which revealed a well-defined and well corticated, globular shaped lesion located on the distal femur, in between the two femoral condyles affecting more than 50% of bone mass, with homogenous content and perilesional osteosclerosis alternating with osteoporotic areas. Dimensions at first measurement were 4.7 cm in diameter with a relatively round shape.

Computer tomography (CT) and Magnetic Resonance Imaging (MRI) scans followed as imaging investigations because the posterior cortex of the femur was damaged and the lesion extended into soft tissues in the popliteal region.

The MRI scan reveals the characteristic ‘penumbra sign’ that can be observed on T1 weighted images (Figure 1A) and consists of a higher signal intensity rim lining the main lesion, which is of a hypo intense signal on T1 weighted images [5,6,7]. The peripheral intensity is due to the vascularized granulation tissue and is characteristic of Brodie’s abscess helping to differentiate the lesion from malignant tumors [5]. The ‘penumbra sign’ presents a sensitivity of 75% and specificity higher than 90% in the diagnosis of subacute osteomyelitis [5], and was an extremely useful imaging tool for reaching the final diagnosis. MRI study is especially important in the diagnosis of osteomyelitis because it can visualize bone edema and soft tissue spread of the disease [6], which is essential for correct pre-operative planning.

A CT scan was also solicitated for pre-operative planning, and revealed the same globular shaped lesion with a well-defined sclerotic cortex and fluid content that damaged the posterior femoral cortex and infiltrated soft tissues in the popliteal region. CT has good spatial resolution and demonstrates clearly the anatomical relationship between areas of infection and major vessels or other important structures. CT has superior bony resolution to MRI and is better for understanding cortical defects, formation of sequestrums, or periosteal reaction [7]. Intramedullary gas is an ancillary sign of osteomyelitis that is also best seen on CT [8].

## 3. Management and Outcome

Surgical treatment was indicated, together with intra-operative culture (Figure 2C) and specific long term antibiotics after surgery. Open debridement was performed (Figure 2A,D) using an antero-lateral approach to the knee with lateral fenestration of the distal femur under fluoroscopy control (Figure 2B) to directly access the cavity of the abscess and posterior approach to the popliteal region for soft tissue debridement. Antiseptic solutions in large quantities and local antibiotics were used for proper local debridement, before drying the cavity to allow for its correct filling with bone inducing cement.

The remaining bone defect was filled with bone inducing cement to offer structural strength until integration and complete healing (Figure 3C). The cement was prepared according to indications, antibiotics were not added in order to avoid altering the cement properties, but local antibiotic was applied in the remaining cavity just before filling it. Post-operative culture test results revealed *Staphylococcus aureus* infection with penicillin resistance, but otherwise sensitive to common antibiotics. A multidisciplinary decision was to proceed with antibiotic treatment using teicoplanin for 6 weeks. With treatment and postoperative care, blood tests had physiological values within a month post-operatively.

Post-operative evolution was favorable, with blood tests reaching normal physiological values over the next three weeks post-operatively. Pain subsided completely 14 days after surgery, and the patient began rehabilitation exercises as soon as possible up to his pain threshold. At his 6-month follow-up, the patient was fully recovered and had returned to normal activity, with a full range of motion and no pain. Imaging examination at the one-year follow-up showed complete fenestration healing and ongoing integration of the bone inducing cement. Clinical examination revealed a normal range of motion with normal scarring and normal activity levels according to the patient.

## 4. Discussion

Correct initial diagnosis is key to optimal management of Brodie’s abscess, and this involves rapid surgical treatment with aggressive debridement and defect repair in order to avoid septic complications (local and systemic) and traumatic complications, such as pathological fractures due to poor resistance of the remaining bone mass surrounding the abscess. Defect reconstruction can be achieved by using both bone graft [9] and bone inducing cement [10,11,12]. In this particular case, the second option was chosen because of the large dimensions of the abscess cavity. Bone inducing cement filled the cavity perfectly and offered structural support post-operatively to avoid pathological fractures following debridement. In this particular case, circumferential cortical bone offered enough support to avoid osteosynthesis. The original and interesting aspect in this specific case is the episodic evolution in the case of 3 years of neglect, with episodes of 4–6 weeks of pain alternating with 6-month long periods of clinical remission. The long-term evolution of this Brodie’s abscess is not seen much nowadays due to good addressability and medical care, but the fact that the patient refused any medical care is what led to this three-year long evolution that offers extremely valuable information on Brodie’s abscess prognosis and evolution in the case of conservative treatment methods.

MRI is the best investigation method for the final diagnosis of osteomyelitis because it demonstrates bone marrow edema, confirms the presence of abscesses, and delineates extraosseous disease spread. If MRI is unavailable or medically contra-indicated, nuclear medicine studies and CT are useful alternatives. CT is limited by its poor soft tissue resolution compared to MRI, and it is unable to demonstrate bone marrow edema, which means that a normal CT aspect does not exclude early osteomyelitis. Other disadvantages of computer tomography include patient radiation and metallic artefacts [6]. Despite these limitations, CT remains a useful alternative when MRI is unavailable or contraindicated.

## 5. Conclusions

Brodie’s abscess does not evolve into a malignant, septic condition even if it evolves naturally with complications such as fistulation.

Single-step aggressive debridement and defect reconstruction using bone inducing cement, combined with targeted systemic and local antibiotherapy is an effective treatment management plan that leads to full recovery.

The penumbra sign is a reliable MRI imaging characteristic for the final diagnosis of Brodie’s abscess. MRI study is the best imaging method to appreciate the extent of soft tissue spread of infection or to reveal bone edema in early stages of the infection [13].

## Figures and Tables

**Figure 1 medicina-57-00544-f001:**
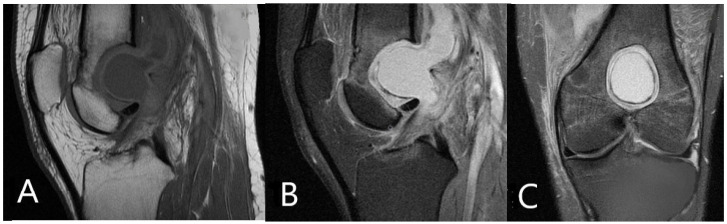
(**A**) T1 weighted MRI image demonstrating ‘penumbra sign’ described as perilesional lining of higher signal intensity, with a central hypo intense content. (**B**) Sagital plane image in fat suppression incidence showing extent of bone mass damage. (**C**) Frontal plane image also in fat suppression incidence demonstrating bone mass damage.

**Figure 2 medicina-57-00544-f002:**
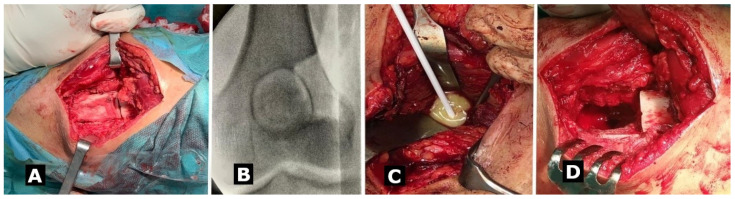
(**A**) Anterolateral approach with bone fenestration of the distal femur to access the cavity of the abscess. (**B**) Bone window placed under fluoroscopy control. (**C**) Intraoperative culture test was harvested. (**D**) Intraoperative aspect after physicochemical debridement using pressure lavage and antiseptic solutions.

**Figure 3 medicina-57-00544-f003:**
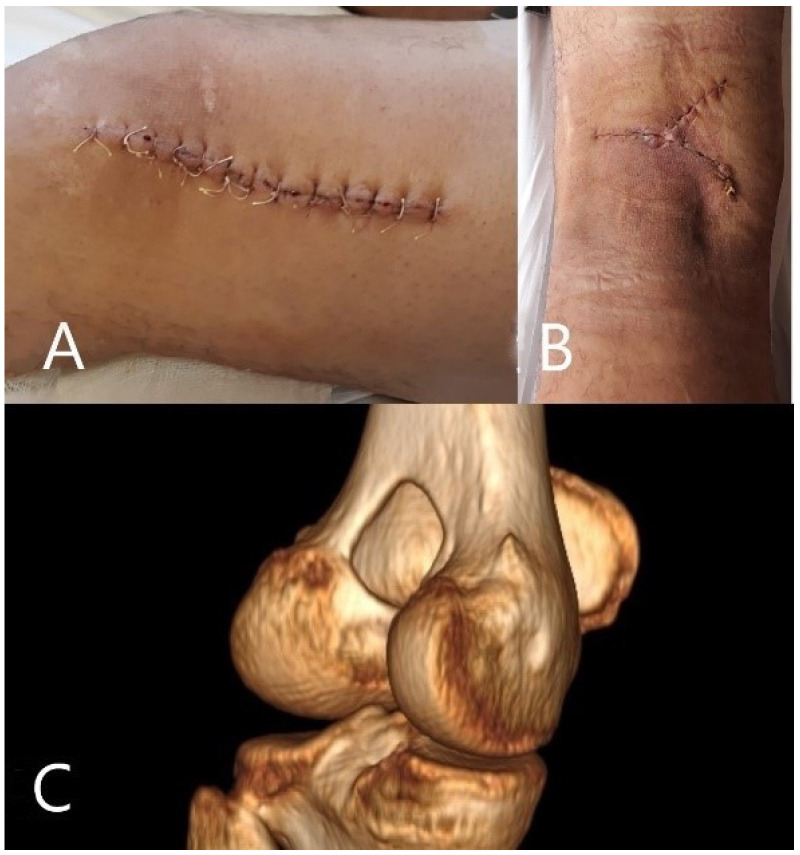
(**A**) Anterolateral incision with normal healing and no sign of septic complications. (**B**) Posterior scar used for soft tissue debridement. (**C**) CT reconstruction after surgery showing post-operative aspect with bone inducing cement filling the remaining bone defect.

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
