# Peer review of "Rare Intercondylar Distal Femoral Brodie’s Abscess in a 21-Year-Old Man Who Refused Medical Care for Three Years after Initial Symptoms"

_medicina, 2021, doi:10.3390/medicina57060544_

Round 1
Reviewer 1 Report
The article is great and very interesting. The topic is well explained, I suggest you to amplify the references, for example by explaining the specificity of RMN for abscess as point out in this article "A rare case of Spinal Epidural Abscess following mesotherapy: a challenging diagnosis and the importance of clinical risk management." , or talking about new markers that could lead to an early diagnosis of a chronic septic state (es: " An immunohistochemical study of the diagnostic value of TREM-1 as marker for fatal sepsis cases").
Good luck!
Author Response
Thankyou very much for your time and the clear observations,
I have added a few more refferences in order to better explain the role of each imaging investigation in osteomyelitis in general, and have added these explanations in the material and methods and the discussions segments of our manuscript.
With utmost respect
Best regards
Adrian Emil Lazarescu
Reviewer 2 Report
REVIEW COMMENT:
2. Materials and Methods
It would be interesting to know if the patient remembers previous trauma in that area. In fact a hypothesis about the formation of Brodie’s abscess suggests that the predisposition of bone to infection increases after minor traumas (without open wounds) [Van der Naald N, Smeeing DPJ, Houwert RM, Hietbrink F, Govaert GAM, van der Velde D. Brodie’s abscess: A systematic review of reported cases. J Bone Jt Infect. 2019;4:33–9]
3. Management and outcome
it would be recommended to specify whether antibiotic-loaded cement was used.
Author Response
Thankyou sincerely for your time and clear observations
I have added in the anamnestic exam that the patient does not remember any traumatic history and the paper you suggested was first reference in our list.
Also i explained better that we used antibiotics locally (vancomicin) as the last step of the debridement just before cementing. We felt that we should follow the preparation indications exactly and we thought that mixing antibiotics might alter the properties of the bone inducing cement.
Again thankyou for reviewing our paper
With utmost respect
Dr. Lazarescu Adrian Emil